# Co-Immobilization of Glucose Dehydrogenase and Xylose Dehydrogenase as a New Approach for Simultaneous Production of Gluconic and Xylonic Acid

**DOI:** 10.3390/ma12193167

**Published:** 2019-09-27

**Authors:** Jakub Zdarta, Karolina Bachosz, Oliwia Degórska, Agata Zdarta, Ewa Kaczorek, Manuel Pinelo, Anne S. Meyer, Teofil Jesionowski

**Affiliations:** 1Institute of Chemical Technology and Engineering, Faculty of Chemical Technology, Poznan University of Technology, Berdychowo 4, PL-60965 Poznan, Poland; 2Department of Chemical and Biochemical Engineering, DTU Chemical Engineering, Technical University of Denmark, Soltofts Plads 229, DK-2800 Kgs. Lyngby, Denmark; 3Department of Biotechnology and Biomedicine, DTU Bioengineering, Technical University of Denmark, Soltofts Plads 224, DK-2800 Kgs. Lyngby, Denmark

**Keywords:** glucose dehydrogenase, xylose dehydrogenase, enzymes immobilization, co-immobilization, silica SBA 15, biomass conversion

## Abstract

The conversion of biomass components catalyzed via immobilized enzymes is a promising way of obtaining valuable compounds with high efficiency under mild conditions. However, simultaneous transformation of glucose and xylose into gluconic acid and xylonic acid, respectively, is an overlooked research area. Therefore, in this work we have undertaken a study focused on the co-immobilization of glucose dehydrogenase (GDH, EC 1.1.1.118) and xylose dehydrogenase (XDH, EC 1.1.1.175) using mesoporous Santa Barbara Amorphous silica (SBA 15) for the simultaneous production of gluconic acid and xylonic acid. The effective co-immobilization of enzymes onto the surface and into the pores of the silica support was confirmed. A GDH:XDH ratio equal to 1:5 was the most suitable for the conversion of xylose and glucose, as the reaction yield reached over 90% for both monosaccharides after 45 min of the process. Upon co-immobilization, reaction yields exceeding 80% were noticed over wide pH (7–9) and temperature (40–60 °C) ranges. Additionally, the co-immobilized GDH and XDH exhibited a significant enhancement of their thermal, chemical and storage stability. Furthermore, the co-immobilized enzymes are characterized by good reusability, as they facilitated the reaction yields by over 80%, even after 5 consecutive reaction steps.

## 1. Introduction

Over the last decades, the number of porous materials has significantly increased. Due to their various properties, such as a high porosity, defined pore structure and size, as well as numerous functional groups, they find application in many fields of science and industry [1]. Mesoporous silicas, due to their properties, are widely employed in different fields, such as catalysis, adsorption, enzyme immobilization or drug delivery [1,2]. Among others, Santa Barbara Amorphous 15 silica (SBA 15) is a great example of these materials, owing its name to the place where it was discovered. The structure of the SBA 15 mesoporous silica is composed of hexagonal, two-dimensional internal pores with diameters between 5 nm and 30 nm [3]. Due to such a system, SBA 15 silica is characterized by a considerably expanded surface area, which is why it is a good material for the adsorption of various compounds and/or enzyme immobilization [4]. In contrast to other types of silicas, during the adsorption or immobilization of biologically active substances (for example enzymes) using mesoporous SBA 15, the substances are adsorbed inside the pores, resulting in changes in their volume and size, as well as a better protection of the attached molecules against harsh reaction conditions [5].

As has already been mentioned, numerous molecules, including enzymes, can be attached using the mesoporous silica SBA 15. Immobilization is the process leading to the total or partial limitation of the movement of substances or biological materials by binding them to a carrier. There are three main types of immobilization: (i) without a carrier, (ii) inside the carrier or (iii) onto a carrier [6]. By applying the enzyme to the carrier, the scope of its action is usually broadened, the separation of products from the reaction mixture is enhanced, and the reuse as well as biocatalytic productivity of the biomolecules are improved [7]. However, the selection of a support material plays a key role in the immobilization process. A carrier that is ideal for any type of enzyme has not yet been created; however, among other materials, silica-based materials seem to be a suitable enzyme support due to their susceptibility to surface modification, their non-toxicity, as well as their chemical and physical stability. Moreover, silica possesses numerous hydroxyl groups that promote immobilization, by both covalent binding and adsorption [6]. Mesoporous silica materials, embracing SBA 15, have been previously used as supports for the immobilization of enzymes for various applications, including processes of biomass pretreatment and its further conversion. For instance, Xie et al. confirmed that by immobilizing the biomolecules on a carrier such as silica, their catalytic efficiency can be increased. The carrier was completely inert during the transesterification of soybean oil, but enabled the easy removal of the biocatalysts from the system. Furthermore, after five cycles of oil conversion, the SBA 15-based catalytic system showed great stability and activity, as it retained over 90% of its initial activity [8].

Plant-based biomass is of particular interest, mainly due to the fact that it can be an alternative for the fossil fuels platform, for the production of energy or biofuels, such as ethanol [7]. Biomass consists mainly of cellulose, hemicellulose and lignin, compounds which might act as a platform for obtaining valuable chemical substances [9]. Biomass is acquired from products, waste and residues from agricultural and forestry production, as well as from part of the waste originating from industrial processing. However, in order to obtain valuable compounds from biomass, it has to be properly pretreated to release valuable sugars, such as pentoses (mainly xylose) and hexoses (mainly glucose). Various pretreatment and conversion methods have been developed, including mechanical, physical and biological techniques [10]. Nevertheless, due to mild process conditions, the limited requirement for sophisticated apparatuses and high selectivity, biological methods based on enzymes are of particular interest [11]. It should be emphasized that the effectiveness of hydrolysis or the conversion of lignocellulose substrates depends on their effective preparation and the precise selection of the enzyme combination. 

In our previous study, we demonstrated and explored the protocol for the separate immobilization of glucose dehydrogenase and xylose dehydrogenase using nano-SiO_2_ and mesoporous SBA 15 silica supports. It has been established that, among others, immobilized enzymes exhibited a significantly higher activity over a wide temperature and pH range as compared to the free form of the biocatalysts. Furthermore, we have proved that immobilized biocatalysts are capable of an independent and efficient conversion of xylose into xylonic acid and glucose into gluconic acid [12]. Nevertheless, further study concerning the improvement of the process efficiency as well as facilitating its simplicity are still highly required. In addition, we would like to emphasize that the literature data related to the co-immobilization of enzymes for the simultaneous conversion of biomass compounds are limited.

Therefore, we have here undertaken a study related to the co-immobilization of two industrially relevant enzymes—glucose dehydrogenase (GDH) and xylose dehydrogenase (XDH)—using mesoporous SBA 15 silica for the efficient and concurrent conversion of glucose and xylose into gluconic acid (GA) and xylonic acid (XA), respectively. As part of the research, we also determined the optimal GDH:XDH activity ratio, as well as the effect of process duration and various pH and temperature conditions on the simultaneous production yield of GA and XA. Moreover, the parameters that are important from a practical application point of view, such as the stability and reusability of the co-immobilized biocatalysts, have also been investigated in detail.

## 2. Materials and Methods

### 2.1. Chemicals and Reagents

Commercial mesoporous SBA 15 silica (<150 mm particle size) with a hexagonal pore morphology and pore size of up to 20 nm (SBA 15), glucose dehydrogenase from *Pseudomonas* sp. (EC 1.1.1.118) (GDH), D-glucose (Glu), D-xylose (Xyl), gluconic acid sodium salt (GA), xylonic acid lithium salt (XA), acetate buffer, phosphate buffer and tris-HCl buffer were provided by Sigma-Aldrich (Steinheim, Germany). *β*-Nicotinamide adenine dinucleotide hydrate (NAD^+^), *β*-nicotinamide adenine dinucleotide, reduced disodium salt hydrate (NADH), Coomassie Brilliant Blue CBB G-250, pyridine, MSTFA, hexane, 96% ethanol, hydrochloric acid and 85% H_3_PO_4_ were purchased from Sigma-Aldrich (Steinheim, Germany). Xylose dehydrogenase (EC 1.1.1.175) (XDH) was provided by Megazyme (Bray, Wicklow, Ireland). All chemicals were of analytical grade and were used as received, without further purification.

### 2.2. Co-Immobilization of GDH and XDH

For the co-immobilization of GDH and XDH, 100 mg of SBA 15 silica was immersed in 1 mL of phosphate buffer at pH 8 containing 0.15 mg (30 U) of glucose dehydrogenase or 150 U of xylose dehydrogenase. To evaluate the most suitable activity ratio of GDH:XDH, the immobilization was carried out using various initial GDH:XDH ratios (1:1, 1:2, 1:5, 1:10). The immobilization was performed via the incubation of the samples for 2 h at 4 °C using an IKA KS 4000i control incubator (IKA Werke GmbH, Staufen im Breisgau, Germany) with mixing at 150 rpm. Afterwards, the obtained biocatalytic systems were centrifuged for 15 min (4000 rpm at a temperature of 4 °C) and used for the glucose and xylose conversion experiments. The amount of co-immobilized enzymes (U/g) was determined spectrophotometrically (Jasco V-750 spectrophotometer, Jasco, Tokyo, Japan) based on the Bradford method [13] and was considered as the difference between the initial enzyme amount and the concentration of the proteins in the supernatant after immobilization, also taking into account the mass of the used support material. Based on these measurements, the immobilization yield was also determined by considering the initial and final concentrations of the enzyme and the volume of the solution before and after immobilization.

### 2.3. Activity and Kinetic Parameters of the Free and Immobilized Enzymes

The activity of the free and co-immobilized glucose dehydrogenase and xylose dehydrogenase was examined based on the spectrophotometric measurements at λ = 340 nm (Jasco V-750 spectrophotometer, Jasco, Tokyo, Japan), carried out based on the model reaction of the reduction of the enzymatic cofactor NAD^+^ to NADH. Briefly, the reaction was performed in 3 mL of phosphate buffer at pH 8 containing 20 mM of NAD^+^, and 100 mM of D-glucose or D-xylose, to which 10 U of GDH and 50 U of XDH (the ADH:XDH ratio was 1:5) of the free or co-immobilized enzymes was added. The reaction mixture was than vigorously shaken on a KS260 Basic incubator (IKA Werke GmbH, Staufen im Breisgau, Germany) at 40 °C for 5 min. One unit of free or co-immobilized enzymes activity was defined as the amount of the biocatalyst required to produce 1 mM of product per minute under the optimal assay conditions. The specific activities of the free and co-immobilized GDH and XDH were expressed in U/mg or U/mL (for XDH) and represent the initial enzyme activity retained per unit mass of enzyme, or per unit mass of enzyme and solid support, respectively.

The kinetic parameters, Michaelis-Menten constant (*K_M_*) and maximum reaction rate (*V_max_*) of the free and co-immobilized XDH and GDH were examined based on the above-presented reaction, using various concentrations of NAD^+^, ranging from 0.01 to 50 mM, by measuring the initial reaction rates. The progress of the reaction was followed spectrophotometrically at λ = 340 nm (Jasco V-750 spectrophotometer, Jasco, Tokyo, Japan). The kinetic parameters were examined using a Hanes-Woolf plot under optimum assay conditions.

### 2.4. Conversion of Glucose to Gluconic Acid and Xylose to Xylonic Acid Catalyzed by Free and Co-Immobilized Enzymes

The simultaneous enzymatic conversion of glucose to gluconic acid and xylose to xylonic acid was performed using an activity ratio of 1:5 of free or co-immobilized GDH:XDH. For this reason, 30 U of GDH and 150 U of XDH of the free or co-immobilized biocatalysts were added to the 1 mL of reaction mixture containing: 55 mM of D-glucose, 345 mM of D-xylose and 100 mM of NAD^+^ in phosphate buffer at pH 8. The reaction was carried out for 60 min at 40 °C with mixing at 100 rpm using an IKA KS 4000i control incubator (IKA Werke GmbH, Staufen im Breisgau, Germany). After 60 min, the process was terminated by adding 1 M HCl, and the obtained samples were analyzed by gas chromatography (GC-MS) to determine the concentration of the gluconic acid, xylonic acid and reaction yield. One unit of enzyme activity (U) of the free and co-immobilized enzymes was defined as the amount of GDH or XDH that produced 1 μmol of gluconic and xylonic acid, respectively, per minute. The yields (%) of the gluconic and xylonic acid were calculated as the ratio of the molar concentration of gluconic and xylonic acid obtained in relation to the initial molar concentration of glucose and xylose, respectively. All of the experiments were performed in triplicate, and error bars were presented as mean values ± standard deviation.

#### 2.4.1. Time Course of the Conversion of Glucose to Gluconic Acid and Xylose to Xylonic Acid 

The effect of the process duration on the changes of the concentration of gluconic acid and xylonic acid was determined based on the above-mentioned reaction performed under optimal process conditions (pH 8, temperature 40 °C) over 60 min. The reaction was followed by collecting samples at every specified period of time, and the production of GA and XA was evaluated using GC-MS measurements.

#### 2.4.2. Effect of pH and Temperature on Enzymatic Conversion of Glucose to Gluconic Acid and Xylose to Xylonic Acid

The effect of the pH and temperature (pH and temperature profiles of co-immobilized GDH and XDH) on the conversion of glucose and xylose was examined based on the above-mentioned reaction over a pH ranging from 5 to 10 (at a temperature of 40 °C) using buffer solutions at the desired pH to adjust the pH and at temperature values ranging from 20 to 70 °C with a step size of 10 °C (at pH 8). After 60 min of the process, the reaction was terminated and the samples were analyzed using GC-MS.

#### 2.4.3. Stability of the Free and Co-Immobilized GDH and XDH

The stability the free and co-immobilized GDH and XDH over time was determined based on the simultaneous catalytic conversion of glucose and xylose, after incubating the samples for 120 min in a phosphate buffer solution at pH 8 at 40 °C. After the specified period of time, the samples were used to catalyze the conversion of glucose into gluconic acid and xylose into xylonic acid, as described in Section 2.3. Furthermore, the samples were subjected to a GC-MS analysis, and, based on the obtained results, the relative activity in terms of the GA and XA production was determined. The initial activity of the free and co-immobilized GDH and XDH was defined as a 100% activity. The inactivation constant (*k_D_*) and enzymes’ half-life values (*t_1/2_*) were determined based on the linear regression slope for ln(RA) vs. time.

#### 2.4.4. Storage Stability and Reusability of the Free and Co-Immobilized GDH and XDH

The storage stability of the free and co-immobilized GDH and XDH, stored in phosphate buffer at pH 8 and 4 °C, was examined based on the above-mentioned reaction of glucose and xylose conversion over 20 days of storage. After the process, the samples were analyzed using GC-MS, and the results were used to calculate the relative activity. For the purposes of the study of the storage stability, the initial value of the activity of the free and co-immobilized GDH and XDH was defined as a 100% activity.

The reusability of the co-immobilized GDH and XDH was determined during the production of GA and XA over ten repeated simultaneous conversion cycles of glucose and xylose carried out under optimal process conditions for 60 min. After each conversion step, the biocatalytic system was centrifuged from the reaction mixture, washed several times using a phosphate buffer at pH 8 and placed into a fresh reaction solution. After the process, the samples were analyzed using GC-MS, and the results were used to calculate the GA and XA yields. 

### 2.5. Analytical Techniques

The morphology of the silica SBA 15 before and after the GDH and XDH co-immobilization was investigated using a transmission electron microscopy (TEM) JEOL JEM-1200EX II (JEOL, Tokyo, Japan) instrument at an accelerating voltage of 80 kV.

In order to identify the functional groups that were present on the surface of the analyzed materials and to confirm the effective enzyme immobilization, Fourier transform infrared spectroscopy (FTIR) was used. The samples were analyzed in the form of KBr pellets formed by mixing 1 mg of the sample with 200 mg of anhydrous potassium bromide, over a wavenumber range of 4000–400 cm^−1^, with a resolution of 1 cm^−1^ using a Bruker Vertex 70 apparatus (Bruker, Billerica, MA, Germany).

The porous structure parameters of the SBA 15 silica before and after immobilization were determined using an ASAP 2020 instrument (Micromeritics Instrument Co., Norcross, GA, USA). The surface area was determined based on the multipoint BET (Brunauer–Emmett–Teller) method, while the mean size and total volume of the pores were examined based on the BJH (Barrett–Joyner–Halenda) algorithm, using data for adsorption under relative pressure (*p/p_0_*) at 77 K. The obtained values are presented as the mean of three measurements, and the equipment accuracy (measured error) does not exceed 0.1% of the obtained values.

For the chromatography analysis, the samples were dried using a SpeedVac concentrator (37 °C, ok. 4.5–5 h). The derivatization was preceded by the modification of the carbonyl groups with an oxime formation (reaction with methoxamine). For that purpose, 50 μL of methoxyamine solution (pyridine 20 mg/mL) was added to the samples and mixed well using a thermomixer (Eppendorf, Hamburg, Germany) for 90 min, at 37 °C and 950 rpm. The oxime formation was followed by the sylilation of the hydroxyl groups using 100 μL MSTFA and incubation for 30 min at 37 °C. The samples were dissolved in 500 μL of hexane, mixed using a vortex, and then 500 μL of the samples were transferred to chromatography vials. The samples were analyzed using a GC-MS/MS chromatograph (Pegasus 4D, GCxGC-TOMFMS, LECO, St Joseph, MI, USA) with a BPX-5 column (28 m, 250 μm, 0.25 μm; SGE Analytical Science, Ringwood, Australia). The analyses were performed with helium as a carrier gas (1 mL/min) for 36 min under the following conditions: 70 °C for 2 min, after which the temperature was increased at 10 °C/min to 300 °C, and the final temperature was maintained for 10 min. The sample volume was 1 μL, and the dosage temperature was set to 250 °C. The temperature of the ion source was equal to 250 °C with an electron energy of 70 eV, and the mass spectra between 33–800 m/z were recorded. The data were analyzed using Chroma TOF-GC software (v4.51.6.0), and quantitative analyses for the ion signals at 217 m/z (xylonic acid) and at 333 and 292 m/z (gluconic acid) were performed for all analyzed compounds. Calibration curves based on the measurement of the analogous samples with the known xylonic acid and gluconic acid concentrations (in the range of 0.1 to 2 µg) were used to determine the contents of the compounds that were analyzed in the samples.

### 2.6. Statistical Analysis

All reactions and measurements were carried out in triplicate, and the presented error bars as well as error values represent the mean values ± standard deviation. Tukey’s test by a one-way ANOVA was performed using SigmaPlot 12 (Systat Software Inc., San Jose, CA, USA). Statistically significant differences were established at the level *p* < 0.05.

## 3. Results

### 3.1. Enzyme Co-Immobilization and Characterization

The characteristic of the SBA 15 silica material before and after co-immobilization and the efficiency of the immobilization were examined based on images obtained using a transmission electron microscope (TEM), changes in the porous structure parameters and Fourier transform infrared spectra (FTIR). The SBA 15 silica exhibited a well-ordered hexagonal array of pores, similar to a honeycomb, with the mesopores characterized by diameters of approx. 20 nm (Figure 1a). As can be observed in Figure 1b, after the co-immobilization of GDH and XDH, the surface of the support material is coated by the layer of the enzymes. Furthermore, it can be observed that after the deposition of enzymes, the pores’ diameters decreased to approx. 13 nm. Based on the FTIR spectrum (Figure 1c) and the presence of the signal at 3400 cm^−1^, which was attributed to stretching vibrations of –OH groups, it is clear that hydroxyl groups are present on the silica surface. In addition, signals at a wavenumber range between 1050 and 460 cm^−1^, characteristic for stretching and bending vibrations of ≡Si–O bonds, can be observed. After the co-immobilization of GDH and XDH, additional signals with maxima at 3450 cm^−1^ (ν–OH groups), 2950 cm^−1^ (ν C–H bonds in CH_2_ and CH_3_), 1650 and 1545 cm^−1^ (ν amide I and amide II bonds), 1020 cm^−1^ (ν C–O–C bonds) and 650 cm^−1^ (δ C–C bonds) appeared. The results of the analysis of the porous structure parameters of the SBA 15 silica before and after the co-immobilization (Table 1) showed that upon GDH and XDH co-immobilization, the surface area of the support material dropped by around 35 m^2^/g. Furthermore, significantly lower values of the pore size and pore diameters of around 40% were noticed.

To follow the co-immobilization of the GDH and XDH and to examine the changes in the substrate affinity of the biomolecules upon the c0-immobilization, the kinetic parameters of the free and co-immobilized enzymes as well as their specific activity were examined and compared (Table 2). It can be seen that the specific activity of the free GDH and XDH were comparable and reached, respectively, 39.8 U/mg and 43.3 U/mL. Meanwhile, after the immobilization, the specific activity of the co-immobilized GDH was lower and reached 25.9 U/mg. An even greater decrease was observed for the co-immobilized xylose dehydrogenase, whose specific activity values were about two times lower, as compared to the native enzyme, reaching 21.6 U/mg. It has also been found that the Michaelis-Menten constant *(K_M_*) and the maximum reaction velocity rate (*V_max_*) for the free GDH reached, respectively, 23.2 mM and 6.4 U/mg. By contrast, the *K_M_* and *V_max_* of the free XDH were significantly lower and were found to be 0.116 mM and 0.63 U/mL. Nevertheless, the obtained values are still in the range of the values of the kinetic parameters that are characteristic for both of the above-mentioned enzymes. After the co-immobilization, the substrate affinity of both enzymes decreased, as the *K_M_* value increased by around 30–40%, reaching 30.1 mM for GDH and 0.149 mM for XDH. Furthermore, a lower substrate affinity finds its reflection in the lower values of *V_max_* for both co-immobilized enzymes. The maximum velocity rate of the co-immobilized glucose dehydrogenase dropped by around 30%, as compared to the free enzyme, and was found to be 4.6 U/mg, (6.4 U/mg for the free GDH). Meanwhile, a less prominent decrease of the *V_max_* of the co-immobilized xylose dehydrogenase of around 20% was noticed; as for the free XDH and co-immobilized XDH, these values reached, respectively, 0.63 U/mL and 0.48 U/mg. Finally, in the present study, to examine the sorption capacity of the SBA 15 toward enzymes’ co-immobilization, the total amount of the co-immobilized GDH and XDH was also examined, resulting in 1458 U of the co-immobilized enzymes per 1 gram of the silica support. This value corresponds to a total immobilization yield of 81%. 

### 3.2. Enzymes Co-Immobilization and Gluconic Acid and Xylonic Acid Production

#### 3.2.1. Effect of GDH:XDH Ratio on the Production of Gluconic Acid and Xylonic Acid

It is commonly known that xylose dehydrogenase and glucose dehydrogenase are characterized by different catalytic activities and kinetics; hence, in order to achieve a high efficiency of gluconic acid and xylonic acid production, as well as to optimize the required quantity of enzymes, the effect of various GDH:XDH ratios on the conversion yield was examined (Figure 2). Although the highest productivity of gluconic acid (95%) was noticed for a GDH:XDH ratio equal to 1:1 and 1:2, the observed yields of xylonic acid were equal to approx. 50% and 60%, respectively. The maximal concentration of xylonic acid (89%) was obtained when the GDH:XDH ratio was equal to 1:5, and in this case the efficiency of the gluconic acid production was also very high and reached approx. 90%, which was very similar to the case of the above-mentioned ratios. Furthermore, the differences in the productivity of both acids when comparing GDH:XDH ratios equal to 1:5 and 1:10 were insignificant. Therefore, a GDH:XDH ratio of 1:5 was selected as the most suitable due to the fact that a smaller amount of xylose dehydrogenase was used to achieve a high reaction yield.

#### 3.2.2. Time Course of The Production of Gluconic Acid and Xylonic Acid Catalysed by Co-Immobilized GDH and XDH.

The time course of the conversion of glucose and xylose catalyzed by co-immobilized GDH and XDH was investigated in order to examine the optimal reaction time that allows for a high process yield and in order to compare the productivity of the gluconic acid and xylonic acid. As can be seen in Figure 3, irrespective of the analyzed enzyme, the concentration of the product increased gradually during the first 30 min of the process and reached its maximum after 60 min, which for gluconic acid was equal to 51 mM and for xylonic acid was equal to 307 mM. Simultaneously, the concentration of the substrates also decreased gradually at the beginning of the process and reached a plateau after 60 min. After this time, over 90% of the glucose and 85% of the xylose were converted into corresponding acids, and the further prolongation of the reaction time did not result in higher concentrations of the products. Moreover, the rapid action of the co-immobilized enzymes was confirmed by the fact that the concentrations of the products exceeded 50% after 15 min and after 20 min of the process for GDH and XDH, respectively.

#### 3.2.3. Effect of pH and Temperature on The Production of Gluconic and Xylonic Acid Catalyzed by Co-Immobilized GDH and XDH

The effect of the pH and temperature on the production yield of xylonic acid and gluconic acid was studied over a wide range of pHs from 5 to 10 and temperature values ranging from 20 °C to 70 °C in order to evaluate the most suitable reaction conditions for the highest conversion of glucose and xylose. It can be seen that in a slightly acidic and neutral environment (pH 5–8), the productivity of gluconic acid was definitely higher than the productivity of xylonic acid (Figure 4a). Moreover, under acidic conditions, both biocatalysts possessed the lowest activity and the process yield decreased below 50%. The highest reaction yield was observed at pH 8 for both co-immobilized XDH and GDH. Under such pH conditions, the productivity of gluconic acid was equal to 93%, while the xylonic acid yield reached 89%. A high reaction efficiency was also noticed at pH 9, at which the productivity of both acids was equal to approx. 85%. A high gluconic acid yield (90%) was also observed at pH 7; however, at this pH, the productivity of xylonic acid reached 82%. Moreover, the co-immobilized enzymes retained their high catalytic properties even at pH 10, as the yield of both analyzed acids exceeded 80%. Based on Figure 4b, it can be noticed that at temperatures below 40 °C and above 60 °C, the production yield of gluconic acid and xylonic acid was lower than 80%. This was particularly observed at a temperature of 20 °C, at which the conversion of glucose and xylose was lower than 40%. On the other hand, the maximum productivity of gluconic acid and xylonic acid was noticed at temperatures of 40 °C and 50 °C. The yields of gluconic acid and xylonic acid at 40 °C reached 93% and 89%, respectively, while at 50 °C they were equal to 91% and 88%. Moreover, it should be underlined that the efficiency of the catalytic production of GA and XA was above 80%, even when the reaction was carried out at 60 °C. 

#### 3.2.4. Stability of the Free and Co-Immobilized GDH and XDH

The determination of the stability of the co-immobilized enzymes over time under the process conditions is a crucial step that strongly influences the possible practical applications of the obtained biocatalytic systems. It can be seen that, irrespective of the type and form of the enzymes used, a progressive decrease of the catalytic activity over time was observed (Figure 5). However, the attachment of the glucose dehydrogenase to the silica support resulted in an enhancement of its stability. After 120 min of incubation at pH 8 and 40 °C, the co-immobilized GDH retained 80% of its initial activity; meanwhile, the free enzyme maintained less than 40% of its catalytic properties. On the other hand, xylose dehydrogenase was characterized by a higher stability over the analysed time range, as compared to GDH. Nevertheless, after co-immobilization using SBA 15 silica as a support, XDH exhibited approx. 85% of its initial activity, while the free enzyme exhibited less than 50% of its properties after the same period of time. A significant improvement of the stability of the co-immobilized GDH is also reflected by the values of the inactivation constant (*k_D_*) and enzyme half-life (*t_1/2_*). Free glucose dehydrogenase was characterized by *k_D_* and *t_1/2_* equal to 0.0075 1/min and 95.4 min, respectively; meanwhile, for the free XDH, these values were equal to 0.0061 1/min and 115.7 min. As a result of the enzymes’ co-immobilization, the inactivation constant and enzyme half-life were also significantly enhanced. For the co-immobilized GDH, these parameters were 3-fold lower and 2.5-fold higher than those of the free enzyme (0.0025 1/min and 275.5 min), respectively, whereas for the co-immobilized XDH the *k_D_* and *t_1/2_* reached 0.0021 1/min and 323.7 min, which was improved by more than 3-fold when compared to the native biocatalyst. 

#### 3.2.5. Storage Stability and Reusability of the Co-Immobilized GDH and XDH

The storage stability as well as reusability of the co-immobilized GDH and XDH are of particular interest from a practical application point of view. The reusability of the co-immobilized biocatalysts was examined on the basis of the conversion efficiency of xylose and glucose into xylonic acid and gluconic acid over ten consecutive reaction cycles, while the storage stability was determined by analyzing the relative activity of the co-immobilized GDH and XDH over 20 days of storage at 4 °C (Figure 6a,b).

The results of the reusability test (Figure 6a) showed that the productivity of gluconic acid and xylonic acid was almost unaltered over the first five reaction steps and decreased only by approx. 10%. The further reaction yield decreased more significantly and after ten conversion cycles reached approx. 60% and 55%, respectively, for gluconic acid and xylonic acid. The results of the storage stability of the free and co-immobilized GDH and XDH (Figure 6b) indicate that both co-immobilized enzymes were characterized by a better storage stability, as compared to the free enzymes. For the first five days of storage, the catalytic activity of all the tested enzymes remained unaltered and after that started to decrease. Finally, after 20 days of storage, the relative activity of the free GDH was equal to only 49%, while for the immobilized GDH it reached 84%. Furthermore, the co-immobilized XDH was also characterized by a higher relative activity, which was equal to 91%, as the free biocatalyst retained only approx. 65% of its initial properties.

## 4. Discussion

### 4.1. Enzyme Co-Immobilization and Characterization

The morphology and the presence of chemical groups in the structure of the SBA 15 silica before and after immobilization as well as the immobilization effectivity were investigated using transmission electron microscopy and FTIR spectra. Based on the TEM image of the pure silica matrix, it can be observed that this material is characterized by ordered arrays of pores with diameters of approx. 20 nm, which facilitate enzyme binding not only onto the surface of the support but also inside pores. Indeed, after the co-immobilization of biomolecules, a uniform layer of the enzyme molecules can be seen in the TEM image. Furthermore, GDH and XDH molecules are deposited both onto the surface as well as into the pores of the silica material, which enhances the protection of the biomolecules against inactivation and improves their stability [14]. These observations are in agreement with the results of the porous structure analysis, which revealed a significant decrease of the pore size and pore volume, indicating enzyme deposition into the pores of the matrix. The effective co-immobilization of glucose dehydrogenase and xylose dehydrogenase was also indirectly confirmed by the presence of the signals attributed to the –OH, C–H, amide I and amide II, C–O and C–C bands in the FTIR spectrum of the SBA 15 after immobilization. Moreover, slight shifts (by approx. 5 to 10 cm^−1^) of signals attributed to amide I and amide II toward lower wavenumber values were observed. This fact indicates that the biomolecules were bounded by the adsorption immobilization, by the formation of hydrogen bonds and electrostatic interactions between mainly amine groups present in the enzymes’ structure and hydroxyl groups incorporated into the silica structure, as previously reported [15,16]. 

In the study we have also examined the specific activity of the free GDH and XDH and its changes upon co-immobilization. As was expected, after their co-immobilization both enzymes exhibited a lower specific activity. This fact might be related to changes in the microenvironment of the biocatalysts upon immobilization, as well as to the limited access of the substrate molecules to the enzymes’ active sites due to biomolecule deposition into the pores of the silica support [17]. Moreover, as suggested earlier by Sánchez-Moreno, a more prominent decrease of the specific activity of the co-immobilized XDH is probably related to larger changes of the three-dimensional conformation of the amino acid residues in the active site of the XDH, as compared to the GDH [18]. Nevertheless, in the present study, the specific activity of the co-immobilized GDH reached over 25 U/mg, which was significantly higher than the specific activity of the glucose dehydrogenase co-immobilized with amine dehydrogenase by Liu et al. using magnetic nanoparticles, whose specific activity was 18.7 U/mg [19]. These results, together with the results representing the amount of immobilized enzyme (almost 1500 U of the co-immobilized enzymes per 1 gram of the silica SBA 15 support) and high co-immobilization yield, exceeding 80%, clearly show that mesoporous silica is a suitable carrier for enzyme co-immobilization and facilitates the retention of high catalytic properties. By contrast, Delgove et al. co-immobilized cyclohexanone monooxygenase and glucose dehydrogenase on an amino-functionalized agarose-based support. In their study, the total co-immobilization yield reached around 75%, and a significantly lower enzyme loading was noticed, as less than 100 U of the enzymes were co-immobilized on the agarose support [20].

To examine the changes in the substrate affinity of the jointly co-immobilized glucose dehydrogenase and xylose dehydrogenase, the kinetic parameters of the free and immobilized enzymes were examined. It can be seen that the Michaelis-Menten constant of both immobilized dehydrogenases increased upon immobilization, indicating a lower substrate affinity. This fact is directly related to the creation of the diffusional limitations in the transport of the substrates due to the deposition of the biomolecules inside the pores of the support material. Another explanation might be related to the fact that almost 1500 U of the enzyme was co-immobilized, which might lead to the local enzyme overcrowding and blocking the enzymes active sites [21]. Furthermore, as suggested earlier by Baron et al., due to the requirement of the GDH and XDH for cofactor molecules, an additional interaction between the cofactor and support material as well as between the cofactor and substrate molecules cannot be excluded, leading to a lower substrate affinity [22]. Simultaneously, a lower value of the maximum velocity rate was also noticed for both co-immobilized enzymes. Thus, to overcome this limitations and facilitate the reaction rate, a higher substrate concentration should be used in reaction with the co-immobilized enzymes [23]. However, less significant changes in the values of the kinetic parameters were observed in the case of the co-immobilized XDH. These observations are in agreement with our previous study and might be explained by the lower molecular weight and size of the XDH, as compared to the GDH, which reduces the diffusional limitations and enhances the accessibility of the active sites for substrate molecules [24]. Nevertheless, we would like to emphasize that information related to the immobilization of dehydrogenases is very limited; thus, the novelty of the present study is certain.

### 4.2. Enzymes Co-Immobilization and Gluconic Acid and Xylonic Acid Production

It is known that, after the pretreatment, the concentrations of glucose and xylose in the biomass liquors vary depending on the source of the biomass and the pretreatment method; however, the total amount of xylose is usually approx. 5 to even 10 times higher than the glucose content [11,25]. Moreover, both co-immobilized enzymes are characterized by different activities as well as various pH and temperature optima. Thus, it was crucial to find the most suitable process conditions that would facilitate the achievement of a high reaction yield. In the first step of the investigation, the effect of various GDH:XDH ratios was examined. It was established that at a GDH:XDH ratio equal to 1:5, the highest conversion yield (over 90%) of glucose and xylose occurred. Although a higher conversion yield of glucose was noticed at a lower enzyme ratio (1:1; 1:2), a low yield of xylose conversion was also noticed due to the insufficient amount of xylose dehydrogenase in the system. On the other hand, when a GDH:XHD ratio of 1:10 was used, the conversion of monosaccharides decreased. This might be explained by the overcrowding of enzymes, which leads to steric hindrances, blocking the enzyme active sites and, in consequence, lowering the conversion yield, as presented earlier by Zhang et al [26]. A significant effect of the optimal co-immobilized enzymes ratio for effective catalytic action was also observed in our previous study. We found that a co-immobilized xylose dehydrogenase to alcohol dehydrogenase ratio equal to 2:1 was the most effective for the simultaneous conversion of xylose and cofactor regeneration [27]. However, in this study, a GDH:XDH ratio equal to 1:5 was selected as the most suitable and was used in the further steps of the investigation.

It is also known that immobilization may alter the three-dimensional structure of enzymes, leading to changes in their pH and temperature optima. According to our previous study and previously published articles, glucose dehydrogenase from *Pseudomonas* sp. (EC 1.1.1.118) exhibited its highest activity at a pH equal to approx. 8 and at 40 to 45 °C [12,28], while free xylose dehydrogenase possesses its optimum at a pH equal to approx. 8 and at a temperature of 35 °C [12,29]. After enzyme co-immobilization, the optimal conditions for the highest conversion of xylose and glucose were pH 8 and a temperature of 40 °C. Therefore, it could be concluded that the temperature optimum of XDH was slightly shifted toward higher values, probably due to slight changes in the microenvironment of the enzyme active sites upon immobilization [30], while the optimal process conditions for GDH were unaltered after immobilization. It should also be clearly stated that the free enzymes only exhibited a high activity under the above-mentioned conditions, and changes in these parameters resulted in a significant decrease of the conversion yield (data not presented). In contrast, the co-immobilized enzymes were able to achieve an over 80% glucose and xylose conversion yield over a wide range of pHs ranging from 7 to 10 and temperatures ranging from 40 to 60 °C. The significant improvement of the enzyme activity and, in consequence, also of the conversion yield, as compared to the free enzymes, might be explained by several factors. First of all, GDH and XDH are known to be multimeric enzymes that tend to dissociate at temperatures over 50 °C [31]. Using silica-based support materials, multipoint enzyme binding is facilitated, enhancing the rigidity of the biomolecules, stabilizing their structure and preventing the dissociation of subunits. Furthermore, SBA 15 support material provides a protective effect for the biocatalysts against harsh reaction conditions that lead to a significant reduction of the conformational changes of the enzyme structure after immobilization [32]. Our observations stay in agreement with those noticed by Zhuang et al., who used silica SBA 15 for the adsorption immobilization of alkaline protease. The immobilized enzyme exhibited over 20% higher activity recovery over the whole analysed pH range (5–9) [33]. An even better protective effect of mesoporous material on the enzyme activity was presented by Li et al., who synthesized (3-aminopropyl) triethoxysilane (APTES) functionalized mesoporous SBA 15 silica for the covalent immobilization of lipase from *Candida rugosa*. Due to the rigidization of the enzyme structure and protective effect of the silica matrix, the covalently attached enzyme exhibited over 40% higher activity over a temperature range of 40 °C to 70 °C, as compared to the free biocatalyst [34]. Nevertheless, a decrease of the reaction yield at low temperatures and at an acidic pH is probably related to the insufficient thermal activation of the enzyme as well as to the protonation of the ionic groups and side chains of the enzyme, respectively. Due to these facts, electrostatic repulsion occurred, leading to the distortion and/or destruction of the active sites of the biomolecules [35].

Data related to the thermal and chemical stability of the free and co-immobilized GDH and XDH indicate that the stability of the biomolecules was significantly improved after immobilization using SBA 15 silica, as the relative activity of both co-immobilized enzymes was approx. 20% higher compared to free dehydrogenases. These results find a reflection in the values of the enzymes’ half-lives and inactivation constants, which upon immobilization were improved over 3-fold. This is probably related to the fact that upon immobilization an external backbone for the biomolecules was provided, stabilizing the entire enzyme structure due to the creation of the enzyme-support interaction [36]. It has also been previously mentioned that GDH and XDH were immobilized into the pores of the support, providing an additional protection of the biomolecules. Finally, the vibrations of glucose dehydrogenase and xylose dehydrogenase caused by heating were limited, reducing the conformational changes of the enzymes and facilitating the preservation of the proper enzyme shape and properties [37]. The above-mentioned facts hampered the thermal and chemical denaturation of the enzymes, which are the main reasons for the lower relative activity observed for the free enzymes. These findings were also highlighted in another study by Karimi et al., who immobilized trypsin by adsorption using nanostructured mesoporous SBA-15 with compatible pore sizes. The presented results indicated that the adsorbed trypsin retained over 90% of its initial activity after 2 h incubation at 45 °C [38].

The reusability and storage stability are of key importance from the point of view of practical applications of the produced biocatalytic systems. The great advantage of the co-immobilized GDH and XDH is associated with the significant improvement of the above-mentioned features, as they retained over 80% of their initial catalytic properties after five consecutive reaction cycles and after 20 days of storage at 4 °C. The retention of a high activity might be explained by several factors, which also improve the thermal stability of the co-immobilized GDH and XDH. Nevertheless, the fact that dehydrogenases are also co-immobilized into the pores of hexagonal silica, providing protection against inactivation caused by thermal and pH denaturation and limiting the elution of the enzyme from the support, should emphasized. However, a decrease in the conversion yield after several reaction steps is probably related to the inhibition of the biomolecules by the substrates and products, and/or to the enzymes’ inactivation by the repeated use and effect of process conditions [39]. Similar observations have also been reported by Mureşeanu at al. and Wongvitvitchot et al., who immobilized, respectively, laccase from *Trametes versicolor* and cellulase from *Trichoderma reesei* by adsorption using SBA 15 silica as the support. In their studies, however, immobilized enzymes retained approx. 60% of their initial catalytic activity [40,41].

## 5. Conclusions

The main difficulties for the widespread application of enzymatic systems for the conversion of valuable biomass components are related to the high cost of the enzymes and their limited reusability. Therefore, in this study, we present the simultaneous co-immobilization of glucose dehydrogenase and xylose dehydrogenase using mesoporous SBA 15 silica for the concurrent conversion of glucose and xylose into gluconic acid and xylonic acid, respectively. Based on the FTIR and TEM results, it has been confirmed that the enzymes’ molecules have been effectively attached onto the surface of the support as well as into its pores, ensuring the additional protection of the biomolecules against harsh reaction conditions. It has been also established that the optimal GDH:XDH ratio for the highest yield of GA and XA was equal to 1:5. The use of the above-mentioned system resulted in an approx. 90% conversion of both glucose and xylose after just 60 min of the process. Furthermore, the stability of the enzymes was significantly improved upon immobilization compared to the free GDH and XDH, as the co-immobilized biocatalysts exhibited over a 30% higher activity after 20 days of storage and after 2 h of incubation in harsh conditions. Finally, the co-immobilized enzymes showed a great reusability, as after 5 reaction cycles the conversion yield of glucose and xylose reached over 80%. The presented data clearly illustrate the exceptional potential of silica SBA 15 co-immobilized glucose dehydrogenase and xylose dehydrogenase for the efficient simultaneous conversion of glucose and xylose into valuable products. We believe that the present study may further stimulate the development of catalysts based on immobilized enzymes for the conversion of biomass components; however, further studies in this research area are still required.

## Figures and Tables

**Figure 1 materials-12-03167-f001:**
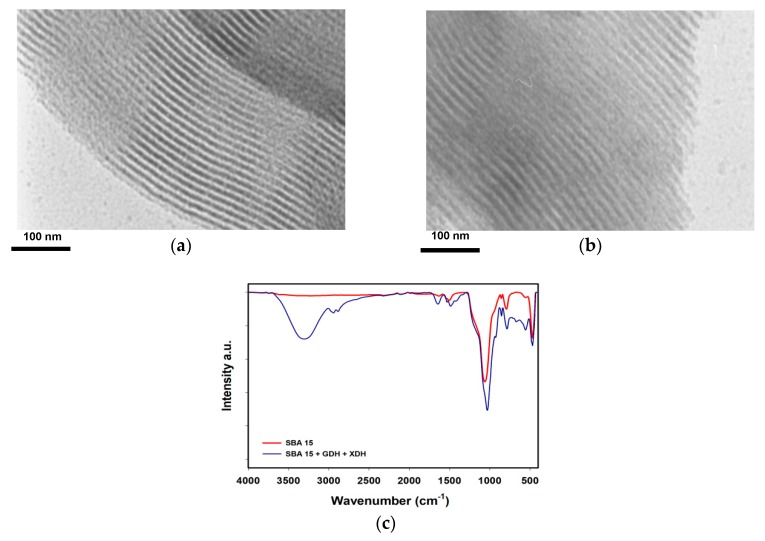
(**a**,**b**) TEM photos and (**c**) FTIR spectra of the SBA 15 silica before and after the GDH and XDH co-immobilization.

**Figure 2 materials-12-03167-f002:**
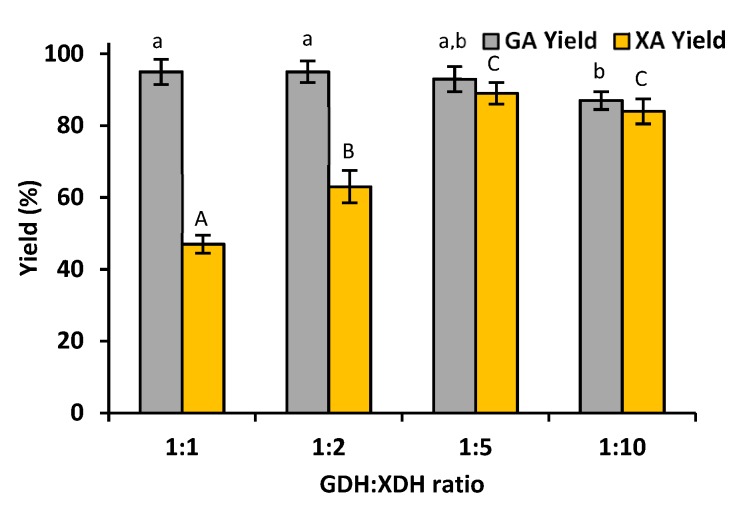
The effect of the ratio of the co-immobilized GDH:XDH on the production of gluconic acid and xylonic acid. The results were analyzed statistically with *p* < 0.05; the lowercase letters refer to statistical differences between the GA yields, while the uppercase letters refer to statistical differences in the XA yields between the different samples.

**Figure 3 materials-12-03167-f003:**
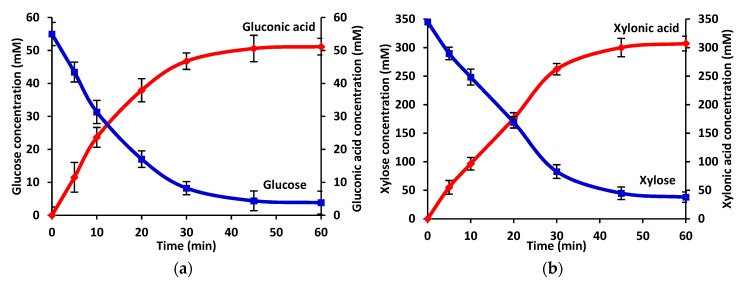
Time course for the reaction of: (**a**) the conversion of glucose into gluconic acid and (**b**) the conversion of xylose into xylonic acid catalyzed by co-immobilized GDH and XDH under optimal process conditions (the blue lines denote the glucose and xylose concentrations, while the red lines denote the concentrations of gluconic and xylonic acid).

**Figure 4 materials-12-03167-f004:**
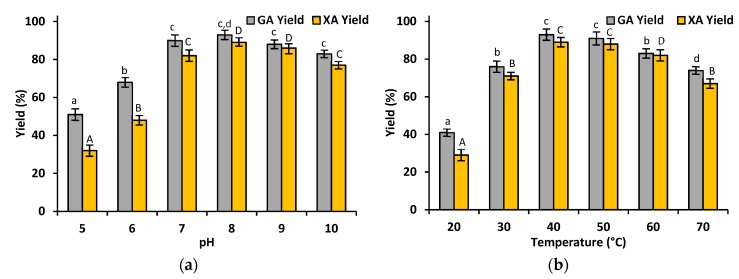
The effect of the (**a**) pH and (**b**) temperature on the conversion of xylose into xylonic acid catalyzed by co-immobilized GDH and XDH. The results were analyzed statistically with *p* < 0.05; the lowercase letters refer to statistical differences between the GA yields, while the uppercase letters refer to statistical differences in the XA yields between the different samples.

**Figure 5 materials-12-03167-f005:**
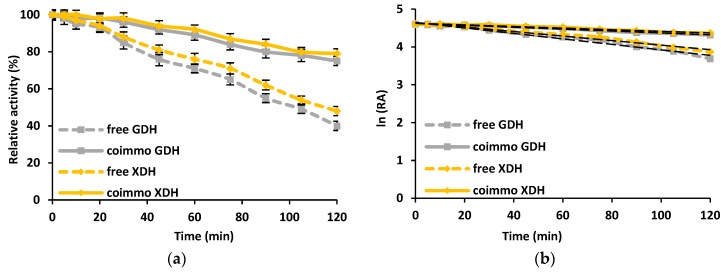
The stability of free and co-immobilized GDH and XDH under optimal process conditions (**a**). The inactivation constants (*k_D_*) were evaluated based on the linear regression slope for free and co-immobilized GDH and XDH under optimal process conditions (**b**).

**Figure 6 materials-12-03167-f006:**
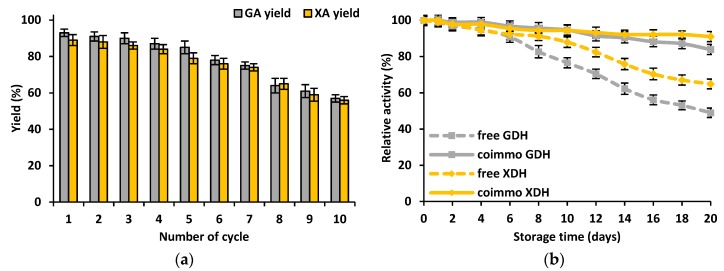
The (**a**) storage stability and (**b**) reusability of the of free and co-immobilized GDH and XDH. The results were analyzed statistically with *p* < 0.05; the lowercase letters refer to statistical differences between the GA yields, while the uppercase letters refer to statistical differences in the XA yields between the different samples.

**Table 1 materials-12-03167-t001:** The porous structure parameters of the SBA15 silica before and after the immobilization.

Sample Name	BET Surface Area (m^2^/g)	Pore Volume (cm^3^/g)	Pore Size (nm)
SBA 15	564.8	0.804	20.362
SBA 15 + GDH + XDH	529.7	0.473	13.573

**Table 2 materials-12-03167-t002:** Data representing the kinetic parameters (Michaelis-Menten constant (*K_M_*) and maximum velocity rate (*V_max_*)) and specific activity of the free and co-immobilized GDH and XDH, as well as the co-immobilization yield and amount of co-immobilized enzymes. * denotes (U/mL).

Sample Name	*K_M_* (mM)	*V_max_* (U/mg)	Specific Activity (U/mg)
free GDH	23.2 ± 1.5	6.4 ± 0.6	39.8 ± 2.3
co-immobilized GDH	30.1 ± 1.6	4.6 ± 0.4	25.9 ± 2.1
free XDH	0.116 ± 0.008	0.63 ± 0.09 *	43.3 ± 1.8 *
co-immobilized XDH	0.149 ± 0.011	0.48 ± 0.07	21.6 ± 2.3

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
