# Peer review of "Co-Immobilization of Glucose Dehydrogenase and Xylose Dehydrogenase as a New Approach for Simultaneous Production of Gluconic and Xylonic Acid"

_materials, 2019, doi:10.3390/ma12193167_

Round 1
Reviewer 1 Report
In this work Zdarta and co-workers present the idea of co-immobilization of two enzymes, glucose dehydrogenase (GDH) and xylose dehydrogenase (XDH), in an inert well characterized and available support, the silica SBA 15, for the simultaneous production of gluconic and xylonic acid from biomass. Although not sufficiently clear from the introduction, it seems that they have already characterized the dual SBA-15 system in a previous work (reference 10 of the manuscript). The authors even used some of the data from their previous work without any proper reference to it i.e. Figure 1 a & b of the manuscript are zooms of Figure 2 a & b of reference 10.
I have also found a very interesting application of a similar double system, GOX/XDH: “Co-immobilization of glucose oxidase and xylose dehydrogenase displayed whole cell on multiwalled carbon nanotube nanocomposite films modified electrode for simultaneous voltammetric detection of D-glucose and D-xylose” (doi. 10.1016/j.bios.2012.10.062) for your consideration.
They concluded that in this article they present the proof-of-concept of the immobilized dual system “…we present the proof-of-concept for the co-immobilization of glucose dehydrogenase and xylose dehydrogenase for using mesoporous SBA 15 silica...” but it is not, they have already published it. Therefore, I am obligated to reject the manuscript or at least to properly adapt the content to summarize their main findings (see below) shortening substantially the manuscript.
Other comments:
- The title should be more accurate. Since the previous characterization of the system was already published last year and because this work is really focused on the optimization of the GDH:XDH ratio, something like “Optimization of the glucose dehydrogenase and xylose dehydrogenase co-immobilized dual system” should be more appropriated.
- The last sentence of the abstract is not very clear (“…indicating possible application”). Please re-write it or even better, remove it.
- The introduction should also be adapted. Please reduce it and account for your previous work highlighting the novel results presented in the current manuscript. Several sections with data and conclusions already presented in your work of 2018 should be removed i.e. Stability of the free and co-immobilized GDH and XDH, etc.
Author Response
Please find the revised version of the manuscript entitled “Co-immobilization of glucose dehydrogenase and xylose dehydrogenase as a new approach for simultaneous production of gluconic and xylonic acid” (Ms. No. Materials-590829), after revision entitled “Co-immobilization of glucose dehydrogenase and xylose dehydrogenase: Optimization and stability study” which we hope after revision is now suitable for publication in Materials Special issue “Silica and Silica-based Materials for Biotechnology, Polymer Composites and Environmental Protection”.
We would like to thank to the Reviewers for insightful review of our work, which contributed to a better understanding of scientific problems relating to the subject of the publication, and will help to eliminate potential errors in the future. Thank you also for the opportunity to re-submit it, incorporating all of the referees’ suggestions. Our comments and changes are noted below, and are marked in yellow in the manuscript.
Reviewer 1
In this work Zdarta and co-workers present the idea of co-immobilization of two enzymes, glucose dehydrogenase (GDH) and xylose dehydrogenase (XDH), in an inert well characterized and available support, the silica SBA 15, for the simultaneous production of gluconic and xylonic acid from biomass.
Query 1: Although not sufficiently clear from the introduction, it seems that they have already characterized the dual SBA-15 system in a previous work (reference 10 of the manuscript). The authors even used some of the data from their previous work without any proper reference to it i.e. Figure 1 a & b of the manuscript are zooms of Figure 2 a & b of reference 10.
Answer 1: We would like to explain that in the previously published paper (Upgrading of Biomass Monosaccharides by Immobilized Glucose Dehydrogenase and Xylose Dehydrogenase) we have presented separate immobilization of GDH and XDH using silica nanoparticles and SBA 15 silica. Meanwhile in this paper we present data about simultaneous co-immobilization of glucose dehydrogenase and xylose dehydrogenase using silica support material. In addition, we would like to explain that TEM photos have not been previously published, nevertheless, to avoid confusions and auto-plagiarism suspicions, in the revised version of the manuscript we have replaced the TEM photos.
Query 2: I have also found a very interesting application of a similar double system, GOX/XDH: “Co-immobilization of glucose oxidase and xylose dehydrogenase displayed whole cell on multiwalled carbon nanotube nanocomposite films modified electrode for simultaneous voltammetric detection of D-glucose and D-xylose” (doi.10.1016/j.bios.2012.10.062) for your consideration.
Answer 2: Thank to the referee for the suggested references. Although it deals with co-immobilization of glucose oxidase (GOX) and xylose dehydrogenase for voltamperometric detection of glucose and xylose (biosensing application), meanwhile in the presented study we have co-immobilized glucose dehydrogenase (GDH) and xylose dehydrogenase for conversion of the above-mentioned compounds into valuable products (biomass conversion application), the suggested references has been added to the references list to better present and support idea of the co-immobilization of the enzymes.
Query 3: They concluded that in this article they present the proof-of-concept of the immobilized dual system “…we present the proof-of-concept for the co-immobilization of glucose dehydrogenase and xylose dehydrogenase for using mesoporous SBA 15 silica...” but it is not, they have already published it. Therefore, I am obligated to reject the manuscript or at least to properly adapt the content to summarize their main findings (see below) shortening substantially the manuscript.
Answer 3: We would like to explain that in 2018 we have published a paper dealing with separate immobilization of glucose dehydrogenase and xylose dehydrogenase using silica support materials. However, in that study, we have immobilized enzymes separately, independently from each other. Meanwhile, in the presented study, we have co-immobilized both enzymes simultaneously, at the same time, using only one support material. Furthermore, in the presented manuscript, the conversion of xylose and glucose occurred concurrent, as in the previously published manuscript, this process was performed separately for each of the compound. In addition, activity, stability and kinetic parameters of the enzymes immobilized separately usually differ from the properties of the enzymes simultaneously co-immobilized. Therefore, we would like to emphasize that although the topic of the study is related to the conversion of biomass components (similar to that published in ChemCatChem in 2018), presented results totally differ from the previously reported and possess significant novelty. Nevertheless, according to the Referee suggestions we have underlined these differences in the Introduction section and we have clearly highlighted the differences between this manuscript and previously published study. In addition, according to the Referees suggestions, the revised version of the manuscript has been improved by addition of the data describing the obtained co-immobilized biocatalytic system. Finally, as suggested by the Referee we have rewritten the indicated sentence to avoid misleading and confusions.
Query 4: The title should be more accurate. Since the previous characterization of the system was already published last year and because this work is really focused on the optimization of the GDH:XDH ratio, something like “Optimization of the glucose dehydrogenase and xylose dehydrogenase co-immobilized dual system” should be more appropriated.
Answer 4: We would like to thank to the Referee for this suggestions. We have changed the title of the manuscript to better highlight the content of the study. The revised version of the manuscript is entitled “Co-immobilization of glucose dehydrogenase and xylose dehydrogenase: Optimization and stability study”
Query 5: The last sentence of the abstract is not very clear (“…indicating possible application”). Please re-write it or even better, remove it.
Answer 5: As suggested by the Referee, the indicated sentence has been rewritten to avoid unnecessary confusions.
Query 6: The introduction should also be adapted. Please reduce it and account for your previous work highlighting the novel results presented in the current manuscript. Several sections with data and conclusions already presented in your work of 2018 should be removed i.e. Stability of the free and co-immobilized GDH and XDH, etc.
Answer 6: We would like to thank to the Reviewer for this comment. According to the Referee suggestion, the Introduction section has been significantly shortened and adapted, to better cover scope of the presented study. In addition, in the revised version of the study, proper explanations of the differences between previously published study and the presented manuscript have been provided. Nevertheless, we would like to explain that presented results concerns co-immobilization of GDH and XDH, meanwhile the manuscript published in 2018 deals wit separate immobilization of GDH and XDH, thus we decided not to remove presented data, as they significantly differ from the results presented in the 2018 manuscript. Furthermore, according to the Referees suggestions, more data about co-immobilization of GDH and XDH has been provided in the revised version of the study.
Reviewer 2 Report
The manuscript “Co-immobilization of glucose dehydrogenase and xylose dehydrogenase as a new approach for simultaneous production of gluconic and xylonic acid” submitted by Jakub Zdarta et al. provides an exhaustive study of the co-immobilization of glucose dehydrogenase (GDH) and xylose dehydrogenase (XDH) onto mesoporous silica SBA-15. In particular, the authors studied the effect of the immobilization on the activity of both enzymes by measuring the production gluconic acid and xylonic acid. Different relevant parameters such as the enzyme ratio, time course for the reaction and the effect of pH and temperature on the enzyme activities were studied. In my opinion, the paper is well-written and supplies relevant information that can be interesting for the field.
However, my main concern is about the novelty of the work. The authors reported a similar data in a recent publication in which they immobilized GDH and XDH in silica nanoparticles (Upgrading of Biomass Monosaccharides by Immobilized Glucose Dehydrogenase and Xylose Dehydrogenase. Jakub Zdarta et al. Chemcatchem. vol. 10. 2019).
Additional concerns and suggestions:
The protein loading and the immobilization efficiency of both proteins should be determined. These results should be discussed in the text and compared with other previous reported systems.
I would suggest also determining specific activities for the immobilized GDH and XDH and compare such activities with the activities of the free enzymes.
The substrate dependence and the kinetic parameters for the immobilized enzymes can be also evaluated and compared with the free enzymes.
Please specify the number of replicates performed and the meaning of the error bars shown in the different graphs.
Related to the previous comment, statistical analysis should be provided in Table 1 and Figures.
Author Response
Please find the revised version of the manuscript entitled “Co-immobilization of glucose dehydrogenase and xylose dehydrogenase as a new approach for simultaneous production of gluconic and xylonic acid” (Ms. No. Materials-590829), after revision entitled “Co-immobilization of glucose dehydrogenase and xylose dehydrogenase: Optimization and stability study” which we hope after revision is now suitable for publication in Materials Special issue “Silica and Silica-based Materials for Biotechnology, Polymer Composites and Environmental Protection”.
We would like to thank to the Reviewers for insightful review of our work, which contributed to a better understanding of scientific problems relating to the subject of the publication, and will help to eliminate potential errors in the future. Thank you also for the opportunity to re-submit it, incorporating all of the referees’ suggestions. Our comments and changes are noted below, and are marked in yellow in the manuscript.
Reviewer 2
The manuscript “Co-immobilization of glucose dehydrogenase and xylose dehydrogenase as a new approach for simultaneous production of gluconic and xylonic acid” submitted by Jakub Zdarta et al. provides an exhaustive study of the co-immobilization of glucose dehydrogenase (GDH) and xylose dehydrogenase (XDH) onto mesoporous silica SBA-15. In particular, the authors studied the effect of the immobilization on the activity of both enzymes by measuring the production gluconic acid and xylonic acid. Different relevant parameters such as the enzyme ratio, time course for the reaction and the effect of pH and temperature on the enzyme activities were studied. In my opinion, the paper is well-written and supplies relevant information that can be interesting for the field.
Query 1: However, my main concern is about the novelty of the work. The authors reported a similar data in a recent publication in which they immobilized GDH and XDH in silica nanoparticles (Upgrading of Biomass Monosaccharides by Immobilized Glucose Dehydrogenase and Xylose Dehydrogenase. Jakub Zdarta et al. Chemcatchem. vol. 10. 2019).
Answer 1: We agree with the Referees that the idea of the enzymes co-immobilization for conversion of selected biomass components has been previously reported. We also agree that in 2018 we have published a paper dealing with separate immobilization of glucose dehydrogenase and xylose dehydrogenase using silica support materials. However, in that study, we have immobilized enzymes separately, independently from each other. Meanwhile, in the presented study, we have co-immobilized both enzymes simultaneously, at the same time, using only one support material. In addition, activity, stability and kinetic parameters of the enzymes immobilized separately usually differ from the properties of the enzymes simultaneously co-immobilized. Furthermore, in the presented manuscript, the co-immobilized enzymes allow simultaneous conversion of xylose and glucose, as in the previously published manuscript, this process was performed separately for each of the compound. Therefore, we would like to emphasize that although the topic of the study is related to the conversion of biomass components (similar to that published in ChemCatChem in 2018), presented results totally differ from the previously reported and possess significant novelty. Finally, although analytical techniques used in this study have also been previously applied, obtained data have not been previously reported. We have underlined these differences in the Introduction section and we have clearly highlighted the differences between this manuscript and previously published study.
Query 2: The protein loading and the immobilization efficiency of both proteins should be determined. These results should be discussed in the text and compared with other previous reported systems.
Answer 2: We would like to thank to the Referee for this valuable comment. According to the Referee suggestion, amount of the co-immobilized enzymes as well as immobilization yield have been determined and are presented and discussed with the previously published study in the revision version of the manuscript in Section 3.1 and Section 4.1.
Query 3: I would suggest also determining specific activities for the immobilized GDH and XDH and compare such activities with the activities of the free enzymes.
Answer 3: The specific activity of the free and co-immobilized enzymes have been determined and compared in the revised version of the manuscript. Obtained data are presented in Table 2 and discussed in the Section 4.2.
Query 4: The substrate dependence and the kinetic parameters for the immobilized enzymes can be also evaluated and compared with the free enzymes.
Answer 4: According to the Reviewer suggestion, the kinetic parameters of the free and co-immobilized enzymes (Michaelis-Menten constant and maximum reaction velocity) have been determined and compared. Obtained results has been also explained and discussed in the Section 4.1 of the revised version of the manuscript.
Query 5: Please specify the number of replicates performed and the meaning of the error bars shown in the different graphs.
Answer 5: We would like to explain that all of the reactions as well as measurement have been carried out in triplicates and the presented results represent the mean values ± standard deviation (error bars) with the exception of the porous structure analysis, which error value reached 0.1% of the measured value and this value is provided by the manufacturer. All of these information have been added in the revised version of the manuscript and marked in yellow.
Query 6: Related to the previous comment, statistical analysis should be provided in Table 1 and Figures.
Answer 6: Thank you for this comment. Statistical analysis of the data presented in Tables and Figures has been provided in the revised version of the manuscript.
Round 2
Reviewer 1 Report
No more comments.
Reviewer 2 Report
The authors addressed al the major and minor drawbacks present in the first version of the manuscript.